# Cytotoxic Activity of Wild Plant and Callus Extracts of *Ageratina pichinchensis* and 2,3-Dihydrobenzofuran Isolated from a Callus Culture

**DOI:** 10.3390/ph16101400

**Published:** 2023-10-03

**Authors:** Mariana Sánchez-Ramos, José Guillermo Encarnación-García, Silvia Marquina-Bahena, Jessica Nayelli Sánchez-Carranza, Antonio Bernabé-Antonio, Valeri Domínguez-Villegas, Emmanuel Cabañas-García, Francisco Cruz-Sosa

**Affiliations:** 1Departament of Biotechnology Autonomous Metropolitan University-Iztapalapa Campus, Av. Ferrocarril de San Rafael Atlixco 186, Col. Leyes de Reforma 1ª. Sección, Alcaldía Iztapalapa, México City 09310, Mexico; 2Faculty of Chemical Science and Engineering, Autonomous University of the State of Morelos, Av. Universidad 1001, Chamilpa, Cuernavaca 62209, Morelos, Mexico; jose.encarnacion@uaem.edu.mx (J.G.E.-G.); valeri.dominguez@uaem.mx (V.D.-V.); 3Chemical Research Center—IICBA, Autonomous University of the State of Morelos, Av. Universidad 1001, Chamilpa, Cuernavaca 62209, Morelos, Mexico; smarquina@uaem.mx; 4Faculty of Pharmacy, Autonomous University of the State of Morelos, Av. Universidad 1001, Chamilpa, Cuernavaca 62209, Morelos, Mexico; jessica.sanchez@uaem.mx; 5Departament of Wood, Pulp and Paper, University Center of Exact Sciences and Engineering, University of Guadalajara, Km. 15.5 Guadalajara-Nogales, Col. Las Agujas, Zapopan 45200, Jalisco, Mexico; antonio.bantonio@academicos.udg.mx; 6Scientific and Technological Studies Center No. 18, National Polytechnic Institute, Blvd. Del Bote 202 Cerro del Gato, Ejido La Escondida, Co. Ciudad Administrativa, Zacatecas 98160, Zacatecas, Mexico; ecabanas@ipe.mx

**Keywords:** cytotoxic activity, 2,3-dihydrobenzofuran, cervical cancer

## Abstract

*Ageratina pichinchensis* (Kunth) R.M. King & H. Rob. belongs to the Asteraceae family and is a plant native to Mexico to which several biological properties are attributed. In this study, the cytotoxic effect of four extracts from the wild plants and two extracts from *A. pichinchensis* callus culture were evaluated against carcinogenic cell lines including prostate carcinoma, cervical cancer, hepatocellular carcinoma, hepatoma human, lung cancer, and cellular keratinocytes. The extracts were obtained with ethyl acetate and methanol using both leaves and stems or the callus. Only the ethyl acetate extract of the callus culture influenced the cervical cancer cell line (HeLa) with an IC_50_ of 94.79 ± 2.0 µg/mL. From the ethyl acetate callus extract, 2,3-dihydrobenzofuran was isolated and purified and also evaluated against cancer cells. The cytotoxic evaluation of this compound showed a significant effect against the HeLa cell line with an IC_50_ of 23.86 ± 2.5 µg/mL. Our results contribute to the development of biotechnological alternatives and extraction processes to produce compounds with possible potential against certain types of human cancer.

## 1. Introduction

*Ageratina pichinchensis* (Kunth) R.M. King & H. Rob. belongs to the family Asteraceae and is an endemic plant species of America. In the state of Morelos, Mexico, this species grows mainly in the municipalities of Amatlán and Tepoztlán. *A. pichinchensis* is popularly known as “water leaf” and “axihuitl”. Traditionality, this plant is used by communities to treat gastric ulcers, healing, and diseases related to inflammatory events [1]. *A. pichinchensis* is described as a shrub up to 1.5 m tall, stem erect, highly branched, or slightly puberulent [2]. In this regard, scientific evidence dealing with the phytochemical and pharmacological studies has validated its ethnomedical uses [3,4,5,6,7,8]. Moreover, biotechnological studies have shown that *A. pichinchensis* can biosynthesize anti-inflammatory compounds that are structurally similar to those produced by the wild plant. For instance, the production of the compounds 3-epilupeol and a benzofuran was found in callus and cell suspension cultures, which are not produced by wild plants. In fact, the production of these anti-inflammatory compounds could be improved using an airlift reactor, and their production was improved by 1.82- to 1.35-fold, respectively [9,10,11].

Although the phytochemical composition and functional properties of *A. pichinichensis* have been evaluated, other biological properties, such as those related to its cytotoxic effect, have yet to be explored. This topic is relevant due to its impact on health and due to the research for bioactive molecules to treat neoplastic diseases. The WHO estimates that cancer is the leading cause of death in the world and more than 18 million deaths were reported worldwide in 2022 alone, the most common being breast, lung, colon, rectum, and prostate cancers [12,13]. This has led to the search for and development of new solutions based on different approaches, given that the causes are diverse and diagnosis is also challenging; this is the reason why multiple research groups around the world have dedicated themselves to the task of searching for therapeutic compounds of synthetic and natural origin [14,15,16] to be applied in therapies or treatments that are already applied in order to improve their effectiveness. One of the common treatments in cancer patients is chemotherapy, which makes use of molecules that act at different levels of the cell cycle, successfully saving lives or improving the quality of life of patients. Despite the percentage of efficacy of chemotherapies, the search for therapeutic agents continues to be necessary due to the high global demand [17,18,19].

This emphasizes the use of medicinal plants as an important source of compounds with cytotoxic effects in different types of cancer. One of the molecules commonly used in chemotherapies is the tetracyclic diterpenoid compound Taxol^®^, which is isolated from the bark of the genus *Taxus*. This molecule is commercially known as paclitaxel^®^, and due to its molecular complexity, it has been impossible to develop synthetic pathways for its production; this phenomenon has led to its direct extraction from the plant, with the disadvantage that trees require up to 15 years to be able to biosynthesize the compound [20,21]. On the other hand, the vinblastine and vincristine alkaloid-type compounds from *Catharanthus roseus* present yields below 0.001% and their cost is around one billion dollars per kilogram, but they are used for their efficacy in chemotherapies for patients with leukemia [22]; other potentially anticancer alkaloids are colchicine, vindesine, vinorelbine, podophyllotoxin, decotaxel, campotecin, curcumin, apigenin, and vincamine, all of which have a common factor in their origin from plant species [23,24].

On the other hand, extracts of medicinal plants have also shown a cytotoxic effect where the compounds participate synergistically, as has occurred with the species *Aristolochia baetica*, *Artemisia annua*, and *Fagonia indica*, among others [25,26,27]. However, there are numerous plant species in the world that lack studies. In Mexico, more than 300 endemic plants from different families and genera are used, of which only a proportion has been scientifically studied [19,28,29]. On the other hand, there are reports of callus and cells suspension cultures of plants whose main advantage is the production of bioactive compounds in a constant and controlled manner, which is a useful alternative because medicinal plants in wild conditions are affected by different seasonal, environmental, and geographical conditions, among others. These factors are determinant conditions for the constant production and yields of secondary metabolites and put medicinal plants at a disadvantage as a sufficient source for the world’s needs in tumor pathologies.

Therefore, this paper reports the cytotoxic activity of wild plant (Figure 1A) and callus culture (Figure 1B) extracts of *A. pichinchensis* as well as 2,3-dihydrobenzofuran isolated from callus cultures.

## 2. Results and Discussion

### 2.1. Cytotoxic Evaluation of Extracts

We evaluated the cytotoxic activity of the ethyl acetate and methanol extracts obtained from different parts of the wild plant *A. pichinchensis*, including its leaves, stems, and callus cultures. The extracts were evaluated using concentrations of 200, 100, 50, 25, and 12.5 µg/mL against various cell lines (Table 1), including human prostate carcinoma (PC-3), cervical cancer (HeLa), hepatocellular carcinoma (Huh-7), human hepatoma (HepG2) and breast tumor cells (MCF-7). These cell lines represent cell types that, unfortunately, have high incidence and mortality rates globally. In addition, cytotoxicity was evaluated in keratinocytes (HaCat), a human epithelial cell line that exhibits normal differentiation, and was used as a selectivity control [30].

The methanolic and ethyl acetate extracts of the wild plants did not demonstrate a significant cytotoxic effect in any of the cell lines analyzed, even with the maximum concentration evaluated (200 µg/mL).

On the other hand, it was detected that the ethyl acetate extract from wild plant leaves exhibited a mild cytotoxic effect on HeLa and PC-3 cell lines, with mean inhibitory concentration (CI_50_) values of 161.49 µg/mL and 188.6 µg/mL, respectively. By contrast, the methanolic extract from wild plant leaves showed no significant cytotoxic effect on these same cell lines.

Interestingly, although the methanolic extract of stems and leaves did not show a significant cytotoxic effect on any of the cell lines, the methanolic extract of callus cultures did have a cytotoxic effect on the HeLa and PC-3 lines, with IC_50_ values of 150.9 µg/mL and 168.6 µg/mL, respectively. However, the most prominent cytotoxic effect was observed with ethyl acetate extract of callus cultures, especially against the HeLa cell line (IC_50_ = 94.79 µg/mL), followed by PC-3, HepG2, Huh-7, and MCF-7 (IC_50_).

To corroborate the efficacy of our cytotoxic study, we used paclitaxel (PTX) as the reference drug; it is a chemotherapeutic drug widely used in the treatment of various types of cancer, including breast, cervical, prostate, and liver cancers [31,32]. The results reveal a remarkable sensitivity of PXT-treated cell lines with IC_50_ values in the nanomolar (nM) range, underscoring the potential efficacy of this reference drug as an anti-tumor agent in our research context.

These results suggest that ethyl acetate extract of *A. pichinchensis* callus cultures could be a potential interest in future applications relating to the treatment of certain types of cancer, especially in the case of the HeLa cell line. It was therefore important to delve into the composition of the extract and identify if there is any compound to which the activity can be attributed.

### 2.2. Chemical Profile of Ethyl Acetate Extract of Callus Cultures

Since the ethyl acetate extract prepared with callus cultures had an important cytotoxic effect, the chemical content was analyzed by mass gas chromatography coupled with mass spectrometry (Figure 2). The chromatographic and mass spectrometric analysis allowed us to identify 11 main compounds [9].

In addition, according to an NIST (National Institute of Standards and Technology) database search, the identities of the 11 compounds of the structures in Figure 3 are known, and their biological effects have been reported.

Fatty acids (**1**) and (**2**) have been reported as constituents of many plant species such as *Commiphora*, *Orchis chusua*, *Salvia verbenaca*, and *Cleidion javanicum* Bl., and these compounds have shown antioxidant and antimicrobial effects on gram-positive and gram-negative microorganisms, and their antifungal effect has also been reported [33,34,35,36,37]. Particularly, the hexadecenoic acid (**1**) compound has been shown to be an important inhibitor of the PLA2 enzyme that plays a role in inflammation of blood vessels and favors the development of atherosclerosis [38]. However, compound (**2**) has only been associated with antimicrobial effects [39,40]. These compounds are common in the essential oils of many medicinal plants such as *Chasmanthe aethiopica*, *Zostera japonica*, and *Jatropha curcas* [41,42,43], and their effect has been reflected in antimicrobial and anti-inflammatory evaluations.

The compound β-eudesmol acetate (**3**) belongs to the group of eudesmans; it has been reported in species of the Asteraceae family and the main biological activities studied are cytotoxic, antioxidant, and antimicrobial [44,45,46].

Regarding compound (**4**), it has been identified in *Helianthus annuus*, *Achillea millefolium* ssp. *millefolium*, and *Ageratina grandofolia*, whose biological studies showed its antimicrobial and antifungal effect [47,48,49].

On the other hand, the compounds stigmasterol (**5**), stigmasterol glucoside (**11**), and β-sitosterol (**6**) are considered some of the predominant compounds in plants [50,51]. Their pharmacological studies have shown their antimicrobial, anti-inflammatory, anti-osteoarthritic, anti-tumor, cytotoxic, anti-hypercholesterolemia, and antioxidant activities [52,53,54,55].

The 2,3-dihydrobenzofuran compounds have exhibited anti-tumor, antimicrobial, antioxidant, and antiprotozoal activity effects [56,57,58]. Some plant species that biosynthesize 2,3-dihydrobenzofuran are *Aristolochia pubenscens*, *Polygorum barbatum*, and *Tagetespatula* L. [59,60,61]. Compound (**7**) has shown outstanding anti-inflammatory effects and is considered a potential pro-inflammatory factors inhibitor agent [9].

Likewise, the pentacyclic triterpene-type compound β-amyrin (**8**) identified in this study has exhibited anti-inflammatory properties and significantly reduced the expression levels of proinflammatory factors TNF-α, IL-1β, IL-6, PGE-2, and COX-2 [62,63], suggesting that its presence in *A. pichinchensis* extracts may enhance its functional properties by blocking the expression of key enzymes involved in proinflammatory events. In addition, the cytotoxic effect of this metabolite was reported in an MCF-7 breast cancer cell line with IC_50_ 15.5 µg/mL, and it has also shown a cytotoxic effect in colon, ovarian, cervical, lung, and breast cancer lines [64,65]. Moreover, β-amyrin has been reported in species from different families, such as *Alstonia boonei*, *Protium heptaphyllum,* and *Gymnosporia montana* [66,67,68,69,70,71,72]. In addition, antioxidant properties in *Myrcianthes pungens* and *Celastrus hindsii* have been reported [73,74].

On the other hand, the stigmasterol glycoside compound has revealed a cytotoxic effect on kidney, breast, and liver tumor cells [75,76,77], and other reports found that it has anti-inflammatory and antimicrobial effects [78,79,80,81]; finally, the benzofuran compound has been reported as anti-inflammatory [9].

The compound campesterol (**10**) has been reported as an important phagocytosis suppressor and inhibitor of lipopolysaccharide in RAW 264.7 macrophage cells [82]; in addition, extracts of different plant species with anti-inflammatory effects have revealed that compound (**10**) participates in the biological effects attributed to the extracts, such as in *Cajanus cajan* L. seeds, *Ananas comosus* leaves, *Allium schoenoprasum* L. leaves, and in *Opuntia ficus-indica* seed oil [83,84,85,86].

These scientific reports demonstrate that the cytotoxic effect observed in the ethyl acetate extract of *A. pichinchensis* callus in our study is due to biologically active compounds that act synergistically, significantly enhancing its cytotoxic effect on PC-3 and HeLa cell lines. This chemical content is constant, because the callus are cultured under controlled conditions and their nutrients are the same. This advantage has allowed numerous callus cultures to be a source of bioactive compounds, opening up the applications of the cultures in the design of alternative treatments for health problems.

Due the compound 2,3-dihydrobenzofuran having been found to be related to an anti-inflammatory effect with pro-inflammatory potential, it was selected to evaluate its cytotoxic effect; the rest of the compounds have been deeply studied at a pharmacological level, and according to the literature, 2,3-dihydrobenzofurans commonly exhibits cytotoxic, antiviral, antioxidant, antimicrobial, and anti-tumor activities [86,87].

### 2.3. Cytotoxic Effect of the Compound 2,3-Dihydrobenzofuran

Considering the pharmacological properties of 2,3-dihydrobenzofurans, we analyzed the cytotoxic effect of compound (**7**) on the study cell lines. The results show a remarkable effect on the inhibition of HeLa cells, while the rest of the cell lines revealed no inhibitory effect (Figure 4).

It is notable that HeLa cervical cancer cells were the most sensitive to the treatment using 2,3-dihydrobenzofuran (**7**) (Table 2) with an IC_50_ of 23.86 µg/mL, which is 3.97 times lower than the IC_50_ of ethyl acetate extract of callus cultures of *A. pichinchensis*. This may suggest that the cytotoxic activity observed in the extract is associated with the effect of this compound; selectivity against cancer cells was also observed. The selectivity index is commonly reported in the literature as a ratio of IC_50_ values calculated for healthy cells and cancer cells [30,88], with values greater than 1 indicating desirable selectivity against cancer cells. The selectivity index of 2,3-dihydrobenzofuran (**7**) was 5.17, 1.53, 1.54, 1.65, and 1.15 for HeLa, PC-3, Huh-7, HepG2, and MCF-7 cells, respectively.

This finding is outstanding because compound (**7**) is only biosynthesized by callus cultures, but not in the wild plants of *A. pichinchensis*, which provides a notable contribution from callus cultures, as a controlled and stable source of compounds of therapeutic interest. Moreover, compound (**7**) exhibits chirality, which complicates the development of synthesis routes to obtain it. On the other hand, our research group has reported the production of the compound through suspension cell cultures in flasks as well as in airlift reactors, significantly improving its production and possible scaling [10,11], which provides a viable and sustainable alternative to produce the compound.

Methanolic and ethanolic extracts of other wild plant species have reported important cytotoxic effects against HeLa cells, suggesting that there are still plants that can be a source of important cytotoxic compounds; for example, *Artemisia ludoviciana*, *Consolida orientalis* L., *Ferula assa-foetida* L., *Coronilla varia* L., *Moringa oleifera*, and *Ficus carica* L. [89,90,91,92,93]. Likewise, pure compounds have also exhibited a significant inhibitory effect on cell proliferation of the HeLa line, such as the compound benzobijuglone, which was isolated from *Juglans mandshurica*, and the gymnemagenol compound from *Gymnesa sylvestre* [94,95].

Therefore, it is necessary to continue establishing callus cultures of medicinal plants that are a potential source of cytotoxic compounds, and it is even possible to biosynthesize them in species that have not reported a cytotoxic effect, as occurs with *A. pichinchensis*, which is widely used in traditional Mexican medicine. It is used to treat diseases caused by fungal and skin infections and wounds, as well as to relieve pain and treat gastric ulcers, and anti-inflammatory effects have been reported. Extracts from this plant have shown antifungal activity against *Trichophyton mentagrophytes*, *T. rubrum*, and *Candida albicans*, and have shown therapeutic effectiveness in patients with vulvovaginal candidiasis [7,96].

## 3. Materials and Methods

### 3.1. Plant Material from the Wild Plants

The plants were collected in their natural habitat in the San Juan Tlacotenco neighborhood of the municipality of Tepoztlán in the state of Morelos, Mexico. Plants were identified by our research group in previously reported works and assigned the voucher number 39913. The specimen is under protection in the HUMO herbarium of the Autonomous University of the State of Morelos (UAEM), whose taxonomic identification was carried out by Biol. Gabriel Flores Franco [9].

### 3.2. Plant Material from Callus Cultures

Calluses were previously established by our research group using leaf explants in Murashige and Skoog culture medium supplemented with 30 g/L sucrose, 1 mg/L naphthaleneacetic acid, 0.1 mg/L kinetin, and 3 g/L phytagel [8]. The culture medium was sterilized at 121 °C, 15 psi, for 15 min using an autoclave. Calluses were subcultured every 20 days and incubated at 25 ± 2 °C under a photoperiod of 16 h with white fluorescent light (50 µmol/m^2^ s).

### 3.3. Obtaining Organic Extracts

The wild plants were dried at room temperature and the leaves were separated from the stems and ground to a fine powder. For the extraction process, 145.18 g of dried leaves and 78.13 g of dried stems were used. On the other hand, 13.20 g of ground dried calluses were used. The dry plant material (leaves and stems) and dry biomass (callus culture) were extracted by maceration (72 h, at room temperature) with ethyl acetate. The same biomass was then extracted with methanol with three extraction cycles each. The solvent was removed by distillation under reduced pressure, using a rotary evaporator, finally obtaining three ethyl acetate extracts, i.e., leaves (17.20 g), stems (7.31 g), and callus culture (0.924 g), and three methanol extracts, i.e., leaves (59.09 g), stems (28.47 g), and callus culture (2.346 g). This methodology is summarized in Figure 5.

### 3.4. Cytotoxic Assay

PC-3 (prostate), HeLa (cervical), MCF-7 (breast), and Huh-7 and HepG2 (hepatocellular) human cancer cell lines were purchased from ATCC (Manassas, VA, USA) (Figure 5). We also included an immortalized human epidermal keratinocyte line (HaCat) as a control of non-cancerous cells [90]. PC-3 was grown in RPMI-1640 medium (Sigma Aldrich, St. Louis, MO, USA), while Huh-7, HepG2, HeLa, and HaCat were grown in DMEM medium (Invitrogen, Thermo Fisher Scientific, Waltham, MA, USA) supplemented with 10% SFB and 2 mM glutamine. All cultures were incubated at 37 °C in 5% CO_2_.

Cells (4 × 10^3^ cells/well) were seeded in 96-well plates. The cells were treated with the investigated samples (at 200, 100, 50, 25, and 12.5 µg/mL) and incubated at 37 °C in 5% CO_2_ for 48 h. Paclitaxel was used as a positive control. For determining the number of viable cells in proliferation we used a CellTiter 96^®^ AQueous One Solution Cell Proliferation Assay kit (Promega, Madison, WI, USA), following the manufacturer’s instructions. Cell viability was determined by absorbance at 450 nm using an automated ELISA reader (Promega, Madison, WI, USA). Stock solutions of all compounds were prepared in DMSO at a maximum concentration of 0.5%. The experiments were conducted in triplicate in three independent experiments. Data were analyzed using the Prism 8.0 statistical program (Graphpad Software Inc., La Jolla, CA, USA) and the half-maximal inhibitory concentrations (IC_50_) were determined by regression analysis.

### 3.5. Isolation of Compounds from Callus Culture

For the isolation of the compounds, the ethyl acetate callus extract was used. Therefore, we followed the methodology previously reported by our working group [8] and we found high reproducibility. Gas chromatography coupled with mass spectrometry was used for identification of the compounds, and 2 mg of the extract was weighed and dissolved in 1 mL of chloroform to be analyzed by GC-MS (NIST 1.7a). The harvested calluses were dried in an oven at 40 °C, and the dry biomass (14.76 g) was extracted three times with 100 mL of ethyl acetate by maceration; each extraction took place every 72 h at room temperature. The solvent was evaporated under reduced pressure and a yellow viscous extract (0.6859 g) was obtained. The ethyl acetate extract was fractionated on an open chromatographic column pre-packed with 18.76 g of silica gel (70–230 mesh; Merck), and eluted with a gradient system of *n*-hexane/ethyl acetate (90:10, 80:20, 70:30, 60:40, 50:50, 40:60, 30:70, and 00:100 *v*/*v*). Fractions of 10 mL were collected to obtain 45 fractions and they were monitored by TLC (ALUGRAM^®^ SIL G/UV254 silica gel plates). The fractions that showed similarity in TLC were grouped, obtaining 7 groups: GE-1 (1–16; 0.1802 mg), GE-2 (17–18; 15.6 mg), GE-3 (19–20; 78.3 mg), GE-4 (21–25; 40.8 mg), GE-5 (26–36; 81.6 mg), GE-6 (37–40; 70.6 mg), and GE-7 (41–45; 92.6 mg). The GE-1, GE-2, and GE-4 fractions were analyzed by GC-MS; this analysis indicated that the GE-1 fraction was made up of a mixture of *n*-hexadecanoic acid (**1**) and the ester of hexadecanoic acid (**2**). Fraction GE-2 was made up of β-eudesmol acetate (**3**) and desmethoxyencecalin (**4**), and purification of fraction GE-3 by column chromatography using a gradient of *n*-hexane-ethyl acetate (100:00 → 70:30) provided 30 fractions. Fractions 15–22 eluted with *n*-hexane-ethyl acetate (90:10) contained stigmasterol (**5**) and β-sitosterol (**6**), and fractions 23–28 showed a single compound, identified by ^1^H and ^13^C NMR and by comparison with their values reported in the literature [10] as (2S,3R)-5-acetyl-7,3-dihydroxy-2-(1-isopropenyl)-2,3-dihydrobenzofuran (15.3 mg, **7**). The GE-4 fraction was made up of α-amyrin (**8**), 3-epilupeol (**9**), and campesterol (**10**). Compound (**11**) was identified in the GE-5 fraction by direct comparison with an authentic sample available in the laboratory.

## 4. Conclusions

For the first time, it is shown that the chemical content of the ethyl acetate extract of *A. pichinchensis* callus cultures exhibits a significant cytotoxic effect on the HeLa cell line, whose activity is suggested to be attributed to the 2,3-dihydrobenzofuran compound. These results contribute to the development of alternatives for the treatment of cervical cancer, which has become a health problem causing many deaths worldwide; moreover, this finding provides the possibility of new applications of callus culture extracts, since wild plant extracts did not show the cytotoxic effect. This is because in vitro cultures produce the compounds of interest in a constant and controlled manner, while wild plants are dependent on environmental, geographical, seasonal, and other factors.

## Figures and Tables

**Figure 1 pharmaceuticals-16-01400-f001:**
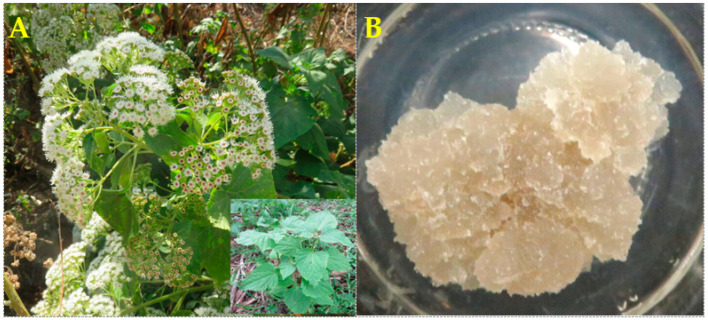
Typical plant of *Ageratina pichinchensis* growing naturally in the state of Morelos, Mexico. Adult plant (**A**); callus induced from leaf explants (**B**).

**Figure 2 pharmaceuticals-16-01400-f002:**
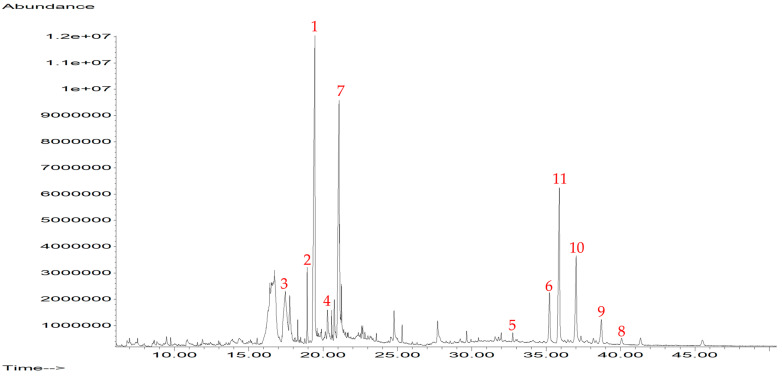
Chromatogram of GC-MS analysis of ethyl acetate callus extract. The red numbers correspond to the compounds identified in the extract.

**Figure 3 pharmaceuticals-16-01400-f003:**
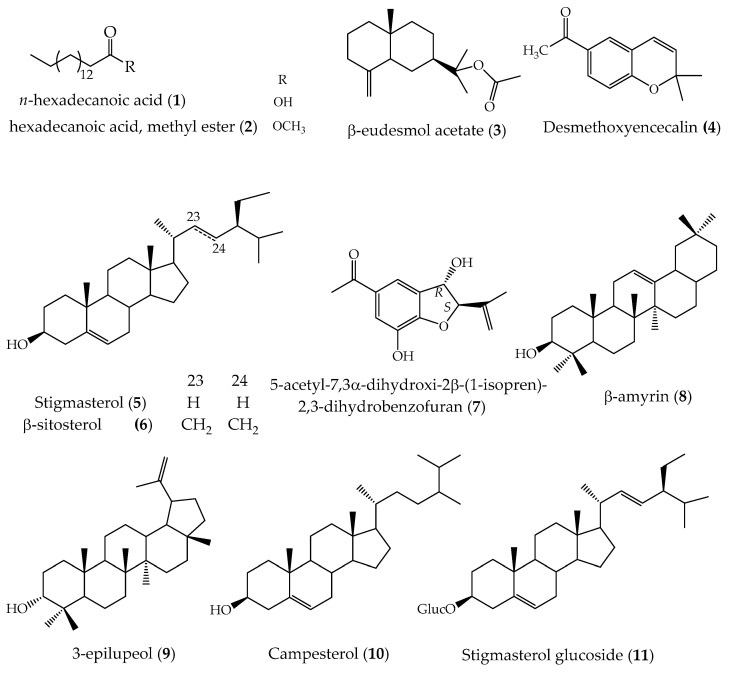
Compounds identified by GS-MS analysis in the ethyl acetate callus extract of *A. pichinchensis*.

**Figure 4 pharmaceuticals-16-01400-f004:**
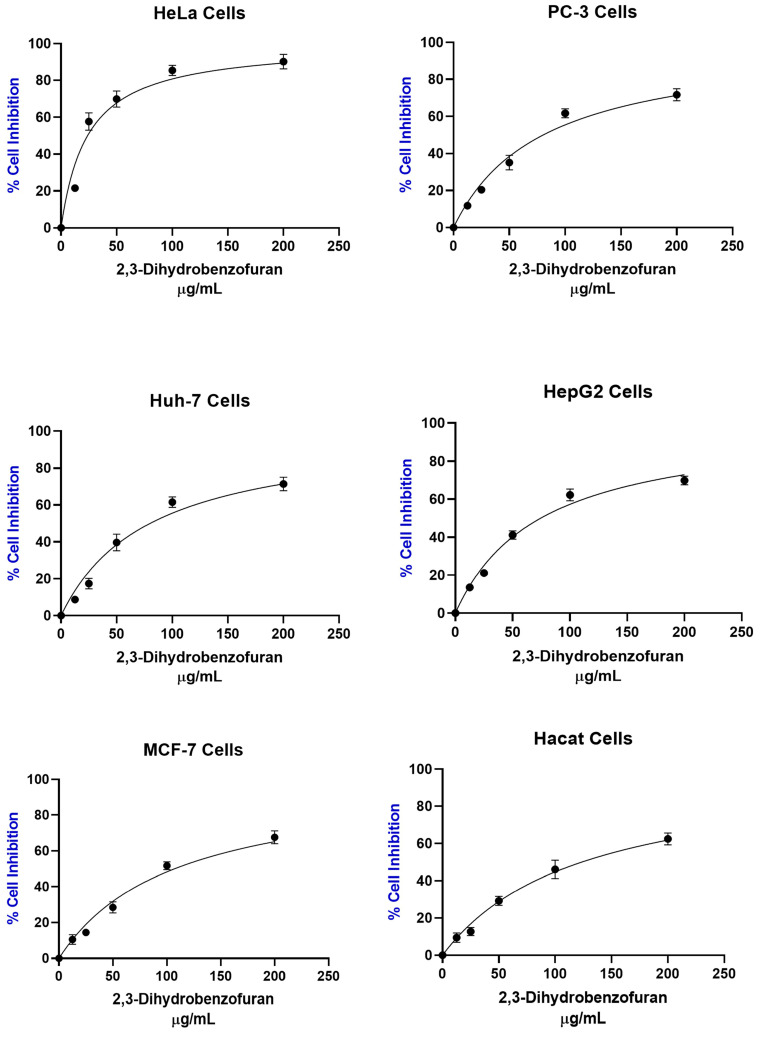
Results of the cytotoxic evaluation of 2,3-dihydrobenzofuran (**7**) against different cancer cell lines.

**Figure 5 pharmaceuticals-16-01400-f005:**
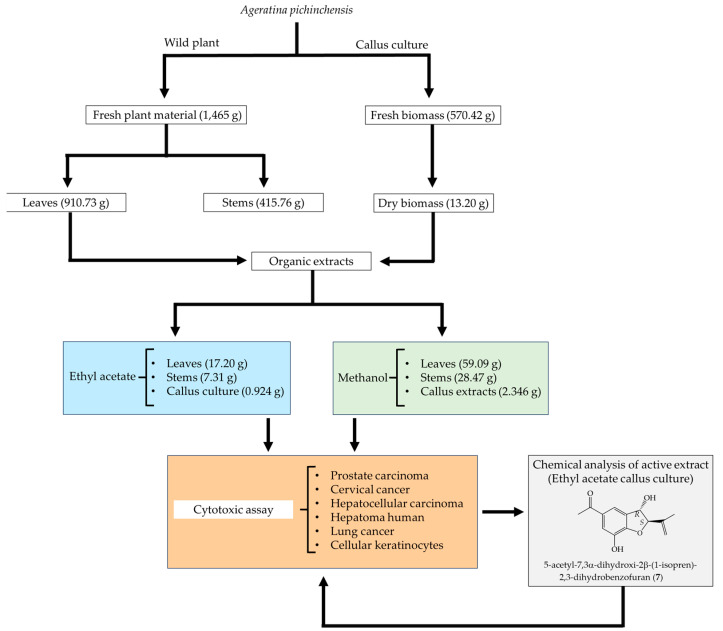
Extraction process and cytotoxic evaluation of *A. pichinchensis*.

**Table 1 pharmaceuticals-16-01400-t001:** Cytotoxic activity (IC_50_) of ethyl acetate and methanol extracts of callus cultures and wild plants of *A. pichinchensis*.

Cellular Line	Wild Plant Leaves		Wild Plant Stems		Callus Cultures		Paclitaxel * (µg/mL)
	EAE (µg/mL)	ME (µg/mL)	EAE (µg/mL)	ME (µg/mL)	EAE (µg/mL)	ME (µg/mL)	
HeLa	161.49 ± 4.91	>200	>200	>200	94.79 ± 2.09	150.9 ± 7.5	0.017 ± 1.03 × 10^−3^
PC-3	188.66 ± 10.42	>200	>200	>200	121.21 ± 9.25	168.6 ± 4.5	0.013 ± 1.96 × 10^−3^
Huh-7	>200	>200	>200	>200	132.80 ± 8.51	>200	0.021 ± 2.73 × 10^−3^
HepG2	>200	>200	>200	>200	122.97 ± 2.54	>200	8.57 × 10^−3^ ± 1.28 × 10^−3^
MCF-7	>200	>200	>200	>200	191.30 ± 6.41	>200	0.018 ± 3.5 × 10^−3^
HaCat	>200	>200	>200	>200	>200	>200	0.097 ± 12.80 × 10^−3^

EAE = ethyl acetate extract; ME = methanol extract. * Reference drug.

**Table 2 pharmaceuticals-16-01400-t002:** Cytotoxic activity of 2,3-dihydrobenzofuran (**7**) against different cancer cell lines.

Cell Cancer Line	IC_50_ (µg/mL)
HeLa	23.86 ± 2.5
PC-3	80.36 ± 4.63
Huh-7	79.98 ± 3.78
HepG2	74.62 ± 2.02
MCF-7	107.20 ± 1.52
HaCat	123.50 ± 15.17

## Data Availability

Not applicable.

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
