# Peer review of "Cytotoxic Activity of Wild Plant and Callus Extracts of Ageratina pichinchensis and 2,3-Dihydrobenzofuran Isolated from a Callus Culture"

_pharmaceuticals, 2023, doi:10.3390/ph16101400_

Round 1
Reviewer 1 Report
The manuscript 'Comparative cytotoxic study of wild plant extracts, their callus cultures and 2,3-dihidrobenzofuran from Ageratina pichinchensis' represents some in vitro investigations. Authors should revise the text.
Main notes:
1. The title of the manuscript should be changed as the authors did not study the isolated compound 2,3-dihidrobenzofuran from Ageratina pichinchens is in the current study, but investigated the chemical content of the ethyl acetate extract from A. pichinchensis (see lines 265-267 of the Conclusions).
2. For better visualization I highly recommend to the authors to prepare Figure 1 in the Introduction section with the whole plant image as well as its callus. It is also worth adding the stages of its extractions and the scheme of anticancer investigations in vitro.
3. In the Abstract and in the Introduction sections it should be mentioned that Ageratina pichinchensis (Kunth) R.M.King & H.Rob. belongs to the Asteraceae family.
4. In the introduction, it is worth giving a brief botanical description of the plant.
5. Line 41: This sentence 'The purification of this extract allowed obtaining a benzofuran type compound as the main product, which was identified as 2,3-dihydrobenzofuran' should be removed from the Abstract as the chemical content of the callus extract was investigated earlier – see Ref. [10].
6. Line 41: This sentence should be redone 'Ageratina pichinchensis is a plants species endemic to the state of Morelos popularly' as this species is not endemic only to the state of Morelos (Mеxico). Please, see below:
https://powo.science.kew.org/taxon/urn:lsid:ipni.org:names:6988-2
7. Lines 41-51: This sentence is too long and should be divided in 2-3.
8. Lines 54-66: This sentence is too long and should be divided in 2-3.
9. Line 272: The term 'in vitro' should be written using italic type.
10. A moderate check is required for the English language and style.
A moderate check is required for the English language and style (first of all, in the Abstract)
Author Response
Responses to Reviewer 1
The manuscript 'Comparative cytotoxic study of wild plant extracts, their callus cultures and 2,3-dihydrobenzofuran from Ageratina pichinchensis represents some in vitro investigations. Authors should revise the text.
Point 1. The title of the manuscript should be chandeg as the authors did not study the isolated compound 2,3-dihydrobenzofuran from Ageratina pichinchensis is in the current study but investigated the chemical content of the ethyl acetate extract from A. pichinchensis (see lines 265-267 of the Conclusions).
Response 1. This is true, in this study 2,3-dihydrobenzofuran was not isolated for the first time in callus cultures, however, to carry out the cytotoxic study we did isolate this compound again from callus culture. Of course, the previously reported methodology was followed. The title has been slightly modified for better understanding.
Point 2. For better visualization I highly recommend to the authors to prepare Figure 1 in the Introduction section with the whole plant image as well as its callus. It is also worth adding the stages of its extractions and the scheme of anticancer investigations in vitro.
Response 2. This was done. Image of plant and callus were added in Introduction section. Moreover, a scheme on extraction and anticancer were added in the Methodology section.
Point 3. In the Abstract and in the Introduction sections it should be mentioned that Ageratina pichinchensis (Kunth) R.M.King & H.Rob. belongs to the Asteraceae family.
Response 3. This was done. Lines 23 and 38.
Point 4. In the introduction, it is worth giving a brief botanical description of the plant.
Response 4. This was done. Lines 43-44.
Point 5. Line 41: This sentence ‘The purification of this extract allowed obtaining a benzofuran type compound as the main producto, which was identified as 2,3-dihydrobenzofuran’ should be removed from the Abstract as the chemical content of the callus extract was investigated earlier -see Ref. [10].
Response 5. This is true, in this study 2,3-dihydrobenzofuran was not isolated for the first time in callus cultures, however, to carry out the cytotoxic study we did isolate this compound again from callus culture in this work. Of course, the previously reported methodology was followed.
Point 6. Line 41: This sentence should be redone ‘Ageratina pichinchensis is a plants species endemic to the state of Morelos popularly’ as this species is not endemic only to the state of Morelos (Mexico). Please, see below: https://powo.science.kew.org/taxon/urn:lsid:ipni.org:names:6988-2.
Response 6. This is true. The text was rewritten. Line 39.
Point 7. Lines 41-51: This sentence is too long and should be divided in 2-3.
Response 7. The text was rewritten. Lines 39-53.
Point 8. Lines 54-66: This sentence is too long and should be divided in 2-3.
Response 8. The text was rewritten. Lines 54-69.
Point 9. Line 272: The term 'in vitro' should be written using italic type.
Response 9. The text was rewritten. Line 353.
Point 10. A moderate check is required for the English language and style.
Response 10. The entire manuscript was reviewed and corrected where necessary.

Reviewer 2 Report
The manuscript “Comparative cytotoxic study of wild plant extracts, their callus cultures and 2,3-dihidrobenzofuran from Ageratina pichinchensis" by Mariana Sánchez‒Ramos et al. is devoted to the study of cytotoxic properties of extracts obtained from Ageratina pichinchensis. At the moment, the manuscript requires a radical revision and may possibly be accepted for reconsideration after major revision.
The authors need to completely revise the abstract they wrote, which currently includes only a brief enumeration of the results obtained. As described in the MDPI journal templates, the typical and most preferred abstract structure should include (1) Background: Place the question addressed in a broad context and highlight the purpose of the study; (2) Methods: briefly describe the main methods or treatments applied; (3) Results: summarize the article's main findings; (4) Conclusions: indicate the main conclusions or interpretations. The abstract should be an objective representation of the article and it must not contain results that are not presented and substantiated in the main text and should not exaggerate the main conclusions. Authors should adhere to this structure.
At the moment, the results presented in subsection “2.1. Cytotoxic evaluation of extracts” are very poor, and therefore the authors need to expand the description of their own results.
What was the reason for the choice of paclitaxel as a positive control? The authors need to add this information to the section “Results and discussion".
Line 101. The authors report that “... ethyl acetate extracts of the wild plants and their callus cultures ...”, however, two lines below, the authors also write “... the ethyl acetate extract of the callus cultures showed a greater effect on the cell lines”. Is this sentence formulated incorrectly? Please correct this misunderstanding or explain the meaning of these sentences.
Based on the data presented in Table 1, is it possible to assume a selective effect of the extracts obtained on a healthy microenvironment, as evidenced by higher IC50 values of the cytotoxic effect on the HaCaT cell line? I suggest that the authors focus on this phenomenon and present its discussion. This should not be limited to just one sentence presented on lines 166-168.
In the title to table 1 or in a footnote under the table, the authors must indicate in what form the results are presented. It is not obvious that the results are presented in the form of IC50 values of the cytotoxic effect.
It is absolutely unclear for what reason in Table 1 in the Paclitaxel column, the authors present almost the results as integers, while the rest are presented up to tenths, which is also incorrect. The authors should have taken a closer look at the presentation of the results and changed it to the standard form, e.g. “1.25±0.06” (both in Table 1 and Table 2).
In Table 1, the results for Paclitaxel are presented in the form of IC50 with a dimension of nM, while for the extracts obtained in mkg/ml. In this regard, for an adequate assessment of the results, it is necessary to recalculate the data for paclitaxel similarly to extracts. At the moment, it is not possible to compare the results obtained.
Were the results for paclitaxel obtained directly by the authors of this manuscript, or are these literary data? In the case of the latter, the authors must provide appropriate references to literary sources. If the results were obtained independently by the authors of this manuscript, it should be noted in the “Materials and methods".
Lines 157-158. The authors report that in this paper they analyzed the effect of 2,3-dihydrobenzofuran on the proliferative activity of cells, but this is not true. The authors should understand the terminology they use, since in this manuscript they present results aimed only at assessing toxic properties, and not antiproliferative action. In order to make such conclusions, the authors need to make at least experiments when evaluating the effect of compounds on cell survival at various time intervals corresponding to cell division cycles. However, it would be better to further evaluate the effect on cellular apoptosis and the molecular mechanisms associated with cell proliferation. At the moment, the experimental part is very poor and requires expansion. A manuscript submitted to a journal of this level cannot be based on a study of cytotoxicity alone.
Lines 174-176. It is absolutely unclear why the authors conclude that their results correlate with the anti-inflammatory activity of 2,3-dihydrobenzofuran. In this paper, the authors were by no means able to identify “a relevant biological activity of the compound". They only evaluated the cytotoxic effect of 2,3-dihydrobenzofuran. In this regard, the sentence on lines 174-176 is incorrect.
Returning to the above remark about the evaluation of antiproliferative properties, the authors should change the title of subsection 3.4.
The authors should change the spelling of the Hacat cell line to HaCaT.
The authors should adhere to the identical spelling of the MCF-7 cell line. On line 100, in table 2, as well as on line 214, this line is written as MCF7, but it is correct to write MCF-7.
Author Response
Reviewer 2
The manuscript “Comparative cytotoxic study of wild plant extracts, their callus cultures and 2,3-dihidrobenzofuran from Ageratina pichinchensis" by Mariana Sánchez‒Ramos et al. is devoted to the study of cytotoxic properties of extracts obtained from Ageratina pichinchensis. At the moment, the manuscript requires a radical revision and may possibly be accepted for reconsideration after major revision.
The authors need to completely revise the abstract they wrote, which currently includes only a brief enumeration of the results obtained. As described in the MDPI journal templates, the typical and most preferred abstract structure should include (1) Background: Place the question addressed in a broad context and highlight the purpose of the study; (2) Methods: briefly describe the main methods or treatments applied; (3) Results: summarize the article's main findings; (4) Conclusions: indicate the main conclusions or interpretations. The abstract should be an objective representation of the article and it must not contain results that are not presented and substantiated in the main text and should not exaggerate the main conclusions. Authors should adhere to this structure.
Point 1. At the moment, the results presented in subsection “2.1. Cytotoxic evaluation of extracts” are very poor, and therefore the authors need to expand the description of their own results.
Response 1. This was done. The description of the results was expanded. Lines 104-142.
Point 2. What was the reason for the choice of paclitaxel as a positive control? The authors need to add this information to the section “Results and discussion".
Response 2. It is well known that paclitaxel is a drug used to treat some types of cancer. Therefore, we use it as a reference control. It is widely used as a positive control in various studies. This was added in the manuscript. Lines 122-127.
Point 3. Line 101. The authors report that “... ethyl acetate extracts of the wild plants and their callus cultures ...”, however, two lines below, the authors also write “... the ethyl acetate extract of the callus cultures showed a greater effect on the cell lines”. Is this sentence formulated incorrectly? Please correct this misunderstanding or explain the meaning of these sentences.
Response 3. This was corrected as suggested. Lines 135, 304.
Point 4. Based on the data presented in Table 1, is it possible to assume a selective effect of the extracts obtained on a healthy microenvironment, as evidenced by higher IC50 values of the cytotoxic effect on the HaCaT cell line? I suggest that the authors focus on this phenomenon and present its discussion. This should not be limited to just one sentence presented on lines 166-168.
Response 4. This was done. The description of the results was expanded and improved. Lines 227-235.
Point 5. In the title to table 1 or in a footnote under the table, the authors must indicate in what form the results are presented. It is not obvious that the results are presented in the form of IC50 values of the cytotoxic effect.
Response 5. This was done. The IC50 was added in the table caption.
Point 6. It is absolutely unclear for what reason in Table 1 in the Paclitaxel column, the authors present almost the results as integers, while the rest are presented up to tenths, which is also incorrect. The authors should have taken a closer look at the presentation of the results and changed it to the standard form, e.g. “1.25±0.06” (both in Table 1 and Table 2).
Response 6. This was done. Table 1 was modified.
Point 7. In Table 1, the results for Paclitaxel are presented in the form of IC50 with a dimension of nM, while for the extracts obtained in mkg/ml. In this regard, for an adequate assessment of the results, it is necessary to recalculate the data for paclitaxel similarly to extracts. At the moment, it is not possible to compare the results obtained.
Response 7. This was done. Table was modified.
Point 8. Were the results for paclitaxel obtained directly by the authors of this manuscript, or are these literary data? In the case of the latter, the authors must provide appropriate references to literary sources. If the results were obtained independently by the authors of this manuscript, it should be noted in the “Materials and methods".
Response 8. The results of paclitaxel were obtained by our experiments. This is described in the methodology. Line 308.
Point 9. Lines 157-158. The authors report that in this paper they analyzed the effect of 2,3-dihydrobenzofuran on the proliferative activity of cells, but this is not true. The authors should understand the terminology they use, since in this manuscript they present results aimed only at assessing toxic properties, and not antiproliferative action. In order to make such conclusions, the authors need to make at least experiments when evaluating the effect of compounds on cell survival at various time intervals corresponding to cell division cycles. However, it would be better to further evaluate the effect on cellular apoptosis and the molecular mechanisms associated with cell proliferation. At the moment, the experimental part is very poor and requires expansion. A manuscript submitted to a journal of this level cannot be based on a study of cytotoxicity alone.
Response 9. Thank you for your comments. It is true that we only evaluate the activity. This has been corrected in the manuscript.
Regarding the comment on the evaluation of apoptosis and mechanism, we inform that our objective for this work was only to evaluate the cytotoxic activity in different cancer lines. We also carried out a comparison between extracts from the wild plant and extracts from callus cultures, which is a biotechnological tool for sustainable use. The most novel thing is that we isolated 2,3-dihydrobenzofuran from callus cultures and for the first time it was evaluated against cancer cell lines, exhibiting good results.
However, in future studies we will be able to evaluate the mechanisms of cell death and the mechanisms of action of the compounds under study.
Point 10. Lines 174-176. It is absolutely unclear why the authors conclude that their results correlate with the anti-inflammatory activity of 2,3-dihydrobenzofuran. In this paper, the authors were by no means able to identify “a relevant biological activity of the compound". They only evaluated the cytotoxic effect of 2,3-dihydrobenzofuran. In this regard, the sentence on lines 174-176 is incorrect.
Response 10. The aim of the study was not to relate the anti-inflammatory effect to the cytotoxic effect. The outstanding finding is the cytotoxic effect of the compound considering that it is a compound only biosynthesized by callus cultures, this contribution is novel because the compound can be scaled up for its massive production in reactors, which suggests a viable alternative for tumor therapies.
Point 11. Returning to the above remark about the evaluation of antiproliferative properties, the authors should change the title of subsection 3.4.
Response 11. This was done. The text was rewritten. Lines 294-313.
Point 12. The authors should change the spelling of the Hacat cell line to HaCaT.
Response 12. This was done. Lines 297, 236 (Table 2), 134 (Table 1), 111,
Point 13. The authors should adhere to the identical spelling of the MCF-7 cell line. On line 100, in table 2, as well as on line 214, this line is written as MCF7, but it is correct to write MCF-7.
Response 13. This was done. Lines 109, 126, 134 (Table 1), 188, 235, 236 (Table 2), 295.

Round 2
Reviewer 1 Report
This manuscript was substantially improved and could be accepted in its present form
Minor editing of English language required
Reviewer 2 Report
The authors have corrected all the shortcomings, thanks! The article may be accepted for publication.